# Predictive-State Decoders:
# Encoding the Future into Recurrent Networks

**Arun Venkatraman**[1*], **Nicholas Rhinehart**[1*], **Wen Sun**[1],
**Lerrel Pinto**[1], **Martial Hebert**[1], **Byron Boots**[2], **Kris M. Kitani**[1], **J. Andrew Bagnell**[1]
[1]The Robotics Institute, Carnegie-Mellon University, Pittsburgh, PA
[2]School of Interactive Computing, Georgia Institute of Technology, Atlanta, GA

## Abstract

Recurrent neural networks (RNNs) are a vital modeling technique that rely on *internal states* learned indirectly by optimization of a supervised, unsupervised, or reinforcement training loss. RNNs are used to model dynamic processes that are characterized by underlying *latent states* whose form is often unknown, precluding its analytic representation inside an RNN. In the Predictive-State Representation (PSR) literature, latent state processes are modeled by an internal state representation that directly models the distribution of future observations, and most recent work in this area has relied on explicitly representing and targeting sufficient statistics of this probability distribution. We seek to combine the advantages of RNNs and PSRs by augmenting existing state-of-the-art recurrent neural networks with PREDICTIVE-STATE DECODERS (PSDs), which add supervision to the network's internal state representation to target predicting future observations. PSDs are simple to implement and easily incorporated into existing training pipelines via additional loss regularization. We demonstrate the effectiveness of PSDs with experimental results in three different domains: probabilistic filtering, Imitation Learning, and Reinforcement Learning. In each, our method improves statistical performance of state-of-the-art recurrent baselines and does so with fewer iterations and less data.

## 1   Introduction

Despite their wide success in a variety of domains, recurrent neural networks (RNNs) are often inhibited by the difficulty of learning an *internal state* representation. Internal state is a unifying characteristic of RNNs, as it serves as an RNN's memory. Learning these internal states is challenging because optimization is guided by the *indirect* signal of the RNN's target task, such as maximizing the cost-to-go for reinforcement learning or maximizing the likelihood of a sequence of words. These target tasks have a *latent state* sequence that characterizes the underlying sequential data-generating process. Unfortunately, most settings do not afford a parametric model of latent state that is available to the learner.

However, recent work has shown that in certain settings, latent states can be characterized by *observations* alone [8, 24, 26] – which are almost always available to recurrent models. In such partially-observable problems (*e.g.* Fig. 1a), a single observation is not guaranteed to contain enough information to fully represent the system's latent state. For example, a single image of a robot is insufficient to characterize its latent velocity and acceleration. While a latent state parametrization may be known in some domains – *e.g.* a simple pendulum can be sufficiently modeled by its angle and angular velocity $(\theta, \dot{\theta})$ – data from most domains cannot be explicitly parametrized.

---

[*]Contributed equally to this work. Direct correspondence to: `{arunvenk,nrhineha}@cs.cmu.edu`

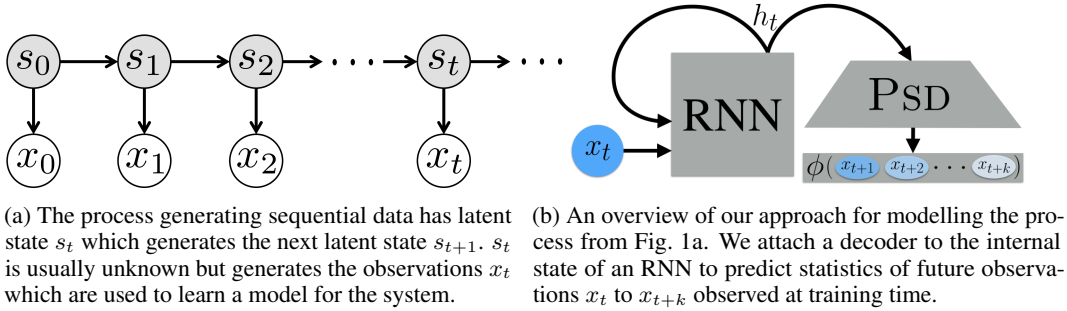

(a) The process generating sequential data has latent state $s_t$ which generates the next latent state $s_{t+1}$. $s_t$ is usually unknown but generates the observations $x_t$ which are used to learn a model for the system.

(b) An overview of our approach for modelling the process from Fig. 1a. We attach a decoder to the internal state of an RNN to predict statistics of future observations $x_t$ to $x_{t+k}$ observed at training time.

Figure 1: Data generation process and proposed model

In lieu of ground truth access to latent states, recurrent neural networks [32, 47] employ internal states to summarize previous data, serving as a learner's memory. We avoid the terminology "hidden state" as it refers to the internal state in the RNN literature but refers to the latent state in the HMM, PSR, and related literature. Internal states are modified towards minimizing the target application's loss, e.g., minimizing observation loss in filtering or cumulative reward in reinforcement learning. The target application's loss is not directly defined over the internal states: they are updated via the chain rule (backpropagation) through the global loss. Although this modeling is indirect, recurrent networks nonetheless can achieve state-of-the-art results on many robotics [18, 23], vision [34, 50], and natural language tasks [15, 20, 38] when training succeeds. However, recurrent model optimization is hampered by two main difficulties: 1) non-convexity, and 2) the loss does not directly encourage the internal state to model the latent state. A poor internal state representation can yield poor task performance, but rarely does the task objective directly measure the quality of the internal state.

Predictive-State Representations (PSRs) [8, 24, 26] offer an alternative internal state representation to that of RNNs in terms of the available observations. Spectral learning methods for PSRs provide theoretical guarantees on discovering the global optimum for the model and internal state parameters under the assumptions of infinite training data and realizability. However, in the non-realizable setting – *i.e.* model mismatch (*e.g.*, using learned parameters of a linear system model for a non-linear system) – these algorithms lose any performance guarantees on using the learned model for the target inference tasks. Extensions to handle nonlinear systems rely on RKHS embeddings [43], which themselves can be computationally infeasible to use with large datasets. Nevertheless, when these models are trainable, they often achieve strong performance [24, 45]; the structure they impose significantly simplifies the learning problem.

We leverage ideas from the both RNN and PSR paradigms, resulting in a marriage of two orthogonal sequential modeling approaches. When training an RNN, PREDICTIVE-STATE DECODERS (Fig. 1b) provide direct supervision on the internal state, aiding the training problem. The proposed method can be viewed as an instance of Multi-Task Learning (MTL) [13] and self-supervision [27], using the inputs to the learner to form a secondary unsupervised objective. Our contribution is a general method that improves performance of learning RNNs for sequential prediction problems. The approach is easy to implement as a regularizer on traditional RNN loss functions with little overhead and can thus be incorporated into a variety of existing recurrent models.

In our experiments, we examine three domains where recurrent models are used to model temporal dependencies: probabilistic filtering, where we predict the future observation given past observations; Imitation Learning, where the learner attempts to mimic an expert's actions; and Reinforcement Learning, where a policy is trained to maximize cumulative reward. We observe that our method improves loss convergence rates and results in higher-quality final objectives in these domains.

## 2 Latent State Space Models

To model sequential prediction problems, it is common to cast the problem into the Markov Process framework. Predictive distributions in this framework satisfy the Markov property:

$$P(s_{t+1}|s_t, s_{t-1}, \ldots, s_0) = P(s_{t+1}|s_t) \tag{1}$$

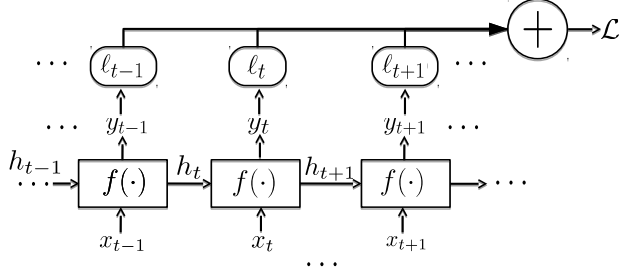

Figure 2: Learning recurrent models consists of learning a function $f$ that updates the internal state $h_t$ given the latest observation $x_t$. The internal state may also be used to predict targets $y_t$, such as control actions for imitation and reinforcement learning. These are then inputs to a loss function $\ell$ which accumulate as the multi-step loss $\mathcal{L}$ over all timesteps.

where $s_t$ is the latent state of the system at timestep $t$. Intuitively, this property tells us that the future $s_{t+1}$ is only dependent on the current state[2] $s_t$ and does not depend on any previous state $s_0, \ldots, s_{t-1}$. As $s_t$ is latent, the learner only has access to observations $x_t$, which are produced by $s_t$. For example, in robotics, $x_t$ may be joint angles from sensors or a scene observed as an image. A common graphical model representation is shown in Fig. 1a.

The machine learning problem is to find a model $f$ that uses the latest observation $x_t$ to recursively update an internal state, denoted $h_t$, illustrated in Fig. 2. **Note that $h_t$ is distinct from $s_t$. $h_t$ is the learner's internal state, and $s_t$ is the underlying configuration of the data-generating Markov Process**. For example, the internal state in the Bayesian filtering/POMDP setup is represented as a belief state [49], a "memory" unit in neural networks, or as a distribution over observations for PSRs.

Unlike traditional supervised machine learning problems, learning models for latent state problems must be accomplished without ground-truth supervision of the internal states themselves. Two distinct paradigms for latent state modeling exist. The first are discriminative approaches based on RNNs, and the second is a set of theoretically well-studied approaches based on Predictive-State Representations. In the following sections we provide a brief overview of each class of approach.

## 2.1 Recurrent Models and RNNs

A classical supervised machine learning approach for learning internal models involves choosing an explicit parametrization for the internal states and assuming ground-truth access to these states and observations at training time [17, 29, 33, 37]. These models focus on learning only the recursive model $f$ in Fig. 2, assuming access to the $s_t$ (Fig. 1a) at training time. Another class of approaches drop the assumption of access to ground truth but still assume a parametrization of the internal state. These models set up a multi-step prediction error and use expectation maximization to alternate between optimizing over the model's parameters and the internal state values [2, 19, 16].

While imposing a fixed representation on the internal state adds structure to the learning problem, it can limit performance. For many problems such as speech recognition [20] or text generation [48], it is difficult to fully represent a latent state inside the model's internal state. Instead, typical machine learning solutions rely on the Recurrent Neural Network architecture. The RNN model (Fig. 2) uses the internal state to make predictions $y_t = f(h_t, x_t)$ and is trained by minimizing a series of loss functions $\ell_t$ over each prediction, as shown in the following optimization problem:

$$\min_f \mathcal{L} = \min_f \sum_t \ell_t(f(h_t, x_t)) \tag{2}$$

The loss functions $\ell_t$ are usually application- and domain-specific. For example, in a probabilistic filtering problem, the objective may be to minimize the negative log-likelihood of the observations [4, 52] or the prediction of the next observation [34]. For imitation learning, this objective function will penalize deviation of the prediction from the expert's action [39], and for policy-gradient reinforcement learning methods, the objective includes the log-likelihood of choosing actions weighted by their observed returns. In general, the task objective optimized by the network does not directly specify a loss directly over the values of the internal state $h_t$.

The general difficulty with the objective in Eq. (2) is that the recurrence with $f$ results in a highly non-convex and difficult optimization [2].

RNN models are thus often trained with backpropagation-through-time (BPTT) [55]. BPTT allows future losses incurred at timestep $t$ to be back-propogated and affect the parameter updates to $f$. These updates to $f$ then change the distribution of internal states computed during the next forward pass through time. The difficulty is then that small updates to $f$ can drastically change the distribution of $h_t$, sometimes resulting in error exponential in the time horizon [53]. This "diffusion problem" can yield an unstable training procedure with exponentially exploding or vanishing gradients [7]. While techniques such as truncated gradients [47] or gradient-clipping [35] can alleviate some of these problems, each of these techniques yields stability by discarding information about how future observations and predictions should backpropagate through the current internal state. A significant innovation in training internal states with long-term dependence was the LSTM [25]. Many variants on LSTMs exist (*e.g.* GRUs [14]), yet in the domains evaluated by Greff et al. [21], none consistently exhibit statistically significant improvements over LSTMs.

In the next section, we discuss a different paradigm for learning temporal models. In contrast with the open-ended internal-state learned by RNNs, Predictive-State methods do not parameterize a specific representation of the internal state but use certain assumptions to construct a mathematical structure in terms of the observations to find a globally optimal representation.

## 2.2 Predictive-State Models

Predictive-State Representations (PSRs) address the problem of finding an internal state by formulating the representation directly in terms of observable quantities. Instead of targeting a prediction loss as with RNNs, PSRs define a belief over the distribution of $k$ future observations, $g_t = [x_t^T, ..., x_{t+k-1}^T]^T \in \mathbb{R}^{kn}$ given all the past observations $p_t = [x_0, \dots x_{t-1}]$ [10]. In the case of linear systems, this $k$ is similar to the rank of the observability matrix [6]. The key assumption in PSRs is that the definition of state is equivalent to having *sufficient* information to predict everything about $g_t$ at time-step $t$ [42], *i.e.* there is a bijective function that maps $P(s_t|p_{t-1})$ – the distribution of latent state given the past – to $P(g_t|p_{t-1})$ – the belief over future observations.

Spectral learning approaches were developed to find an globally optimal internal state representation and the transition model $f$ for these Predictive-State models. In the controls literature, these approaches were developed as subspace identification [51], and in the ML literature as spectral approaches for partially-observed systems [9, 8, 26, 56]. A significant improvement in model learning was developed by Boots et al. [10], Hefny et al. [24], where sufficient feature functions $\phi$ (e.g., moments) map distributions $P(g_t|p_t)$ to points in feature space $\mathbf{E}\left[\phi(g_t)|p_t\right]$. For example, $\mathbf{E}\left[\phi(g_t)|p_t\right] = \mathbf{E}\left[g_t, g_t g_t^T|p_t\right]$ are the sufficient statistics for a Gaussian distribution. With this representation, learning latent state prediction models can be reduced to supervised learning.

Hefny et al. [24] used this along with Instrumental Variable Regression [11] to develop a procedure that, in the limit of infinite data, and under a linear-system realiziablity assumption, would converge to the globally optimal solution. Sun et al. [45] extended this setup to create a practical algorithm, Predictive-State Inference Machines (PSIMs) [44, 45, 54], based on the concept of inference machines [31, 40]. Unlike in Hefny et al. [24], which attempted to find a generative observation model and transition model, PSIMs directly learned the filter function, an operator $f$, that can *deterministically* pass the predictive states forward in time conditioned on the latest observation, by minimizing the following loss over $f$:

$$\ell_p = \sum_t \|\phi(g_{t+1}) - f(h_t, x_t)\|^2, \quad h_{t+1} = f(h_t, x_t) \tag{3}$$

This loss function, which we call the *predictive-state loss*, forms the basis of our PREDICTIVE-STATE DECODERS. By minimizing this supervised loss function, PSIM assigns statistical meaning to internal states: it forces the internal state $h_t$ to match sufficient statistics of future observations $\mathbf{E}\left[\phi(g_t)|p_t\right]$ at every timestep $t$. We observe an empirical sample of the future $g_t = [x_t, \dots, x_{t+k}]$ at each timestep by looking into the future in the training dataset or by waiting for streaming future observations. Whereas [45] primarily studied algorithms for minimizing the predictive-state loss, we adapt it to augment general recurrent models such as LSTMs and for a wider variety of applications such as imitation and reinforcement learning.

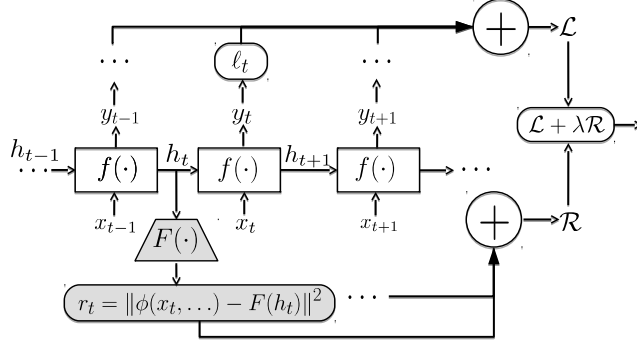

Figure 3: Predictive-State Decoders Architecture. We augment the RNN from Fig. 2 with an additional objective function $\mathcal{R}$ which targets decoding of the internal state through $F$ at each time step to the *predictive-state* which is represented as statistics over the future observations.

## 3 Predictive-State Decoders

Our PREDICTIVE-STATE DECODERS architecture extends the Predictive-State Representation idea to general recurrent architectures. We hypothesize that by encouraging the internal states to encode information sufficient for reconstructing the predictive state, the resulting internal states better capture the underlying dynamics and learning can be improved. The result is a simple-to-implement objective function which is coupled with the existing RNN loss. To represent arbitrary sizes and values of PSRs with a fixed-size internal state in the recurrent network, we attach a decoding module $F(\cdot)$ to the internal states to produce the resulting PSR estimates. Figure 3 illustrates our approach.

Our PSD objective $\mathcal{R}$ is the predictive-state loss:

$$\mathcal{R} = \sum_t \|F(h_t) - \phi([x_{t+1}, x_{t+2}, \ldots])\|_2^2, \quad h_t = f(h_{t-1}, x_{t-1}), \tag{4}$$

where $F$ is a decoder that maps from the internal state $h_t$ to an *empirical sample* of the predictive-state, computed from a sequence of observed future observations available at training. The network is optimized by minimizing the weighted total loss function $\mathcal{L} + \lambda\mathcal{R}$ where $\lambda$ is the weighting on the predictive-state objective $\mathcal{R}$. This penalty encourages the internal states to encode information sufficient for directly predicting sufficient future observations. Unlike more standard regularization techniques, $\mathcal{R}$ does not regularize the parameters of the network but instead regularizes the *output variables*, the internal states predicted by the network.

Our method may be interpreted as an instance of Multi-Task Learning (MTL) [13]. MTL has found use in recent deep neural networks [5, 27, 30]. The idea of MTL is to employ a shared representation to perform complementary or similar tasks. When the learner exhibits good performance on one task, some of its understanding can be transferred to a related task. In our case, forcing RNNs to be able to more explicitly reason about the future they will encounter is an intuitive and general method. Endowing RNNs with a theoretically-motivated representation of the future better enables them to serve their purpose of making sequential predictions, resulting in more effective learning. This difference is pronounced in applications such as imitation and reinforcement learning (Sections 4.2 and 4.3) where the primary objective is to find a control policy to maximize accumulated future reward while receiving only observations from the system. MTL with PSDs supervises the network to predict the future and implicitly the consequences of the learned policy. Finally, our PSD objective can be considered an instance of self-supervision [27] as it uses the inputs to the learner to form a secondary *unsupervised* objective.

As discussed in Section 2.1, the purpose of the internal state in recurrent network models (RNNs, LSTMs, deep, or otherwise) is to capture a quantity similar to that of state. Ideally, the learner would be able to back-propagate through the primary objective function $\mathcal{L}$ and discover the best representation of the latent state of the system towards minimizing the objective. However, as this problem highly non-convex, BPTT often yields a locally-optimal solution in a basin determined by the initialization of the parameters and the dataset. By introducing $\mathcal{R}$, the space of feasible models is reduced. We observe next how this objective leads our method to find better models.

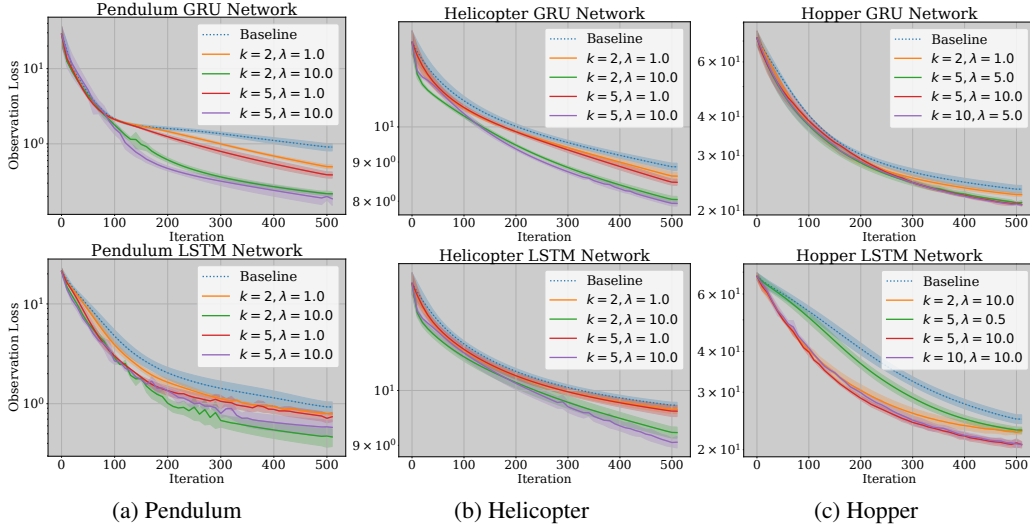

Figure 4: Loss over predicting future observations during filtering. For both RNNs with GRU cells (*top*) and with with LSTM cells (*bottom*), adding PSDs to the RNN networks can often improve performance and convergence rate.

## 4 Experiments

We present results on problems of increasing complexity for recurrent models: probabilistic filtering, Imitation Learning (IL), and Reinforcement Learning (RL). The first is easiest, as the goal is to predict the next future observation given the current observation and internal state. For imitation learning, the recurrent model is given training-time expert guidance with the goal of choosing actions to maximize the sequence of future rewards. Finally, we analyze the challenging domain of reinforcement learning, where the goal is the same as imitation learning but expert guidance is unavailable.

PREDICTIVE-STATE DECODERS require two hyperparameters: $k$, the number of observations to characterize the predictive state and $\lambda$, the regularization trade-off factor. In most cases, we primarily tune $\lambda$, and set $k$ to one of $\{2, \ldots, 10\}$. For each domain, for each $k$, there were $\lambda$ values for which the performance was worse than the baseline. However, for many sets of hyperparameters, the performance exceeded the baselines. Most notably, for many experiments, the convergence rate was significantly better using PSDs, implying that PSDs allows for more efficient data utilization for learning recurrent models.

PSDs also require a specification of two other parameters in the architecture: the featurization function $\phi$ and decoding module $F$. For simplicity, we use an affine function as the decoder $F$ in Eq. (4). The results presented below use an identity featurization $\phi$ for the presented results but include a short discussion of second order featurization. We find that in each domain, we are able to improve the performance of the state-of-the-art baselines. We observe improvements with both GRU and LSTM cells across a range of $k$ and $\lambda$. In IL with PSDs, we come significantly closer and occasionally eclipse the expert's performance, whereas the baselines never do. In our RL experiments, our method achieves statistically significant improvements over the state-of-the-art approach of [18, 41] on the 5 different settings we tested.

### 4.1 Probabilistic Filtering

In the probabilistic filtering problem, the goal is to predict the future from the current internal state. Recurrent models for filtering use a multi-step objective function that maximizes the likelihood of the future observations over the internal states and dynamics model $f$'s parameters. Under a Gaussian assumption (e.g. like a Kalman filter [22]), the equivalent objective that minimizes the negative log-likelihood is given as $\mathcal{L} = \sum_t \|x_{t+1} - f(x_t, h_t)\|^2$.

While traditional methods would explicitly solve for parametric internal states $h_t$ using an EM style approach, we use BPTT to implicitly find an non-parametric internal state. We optimize the

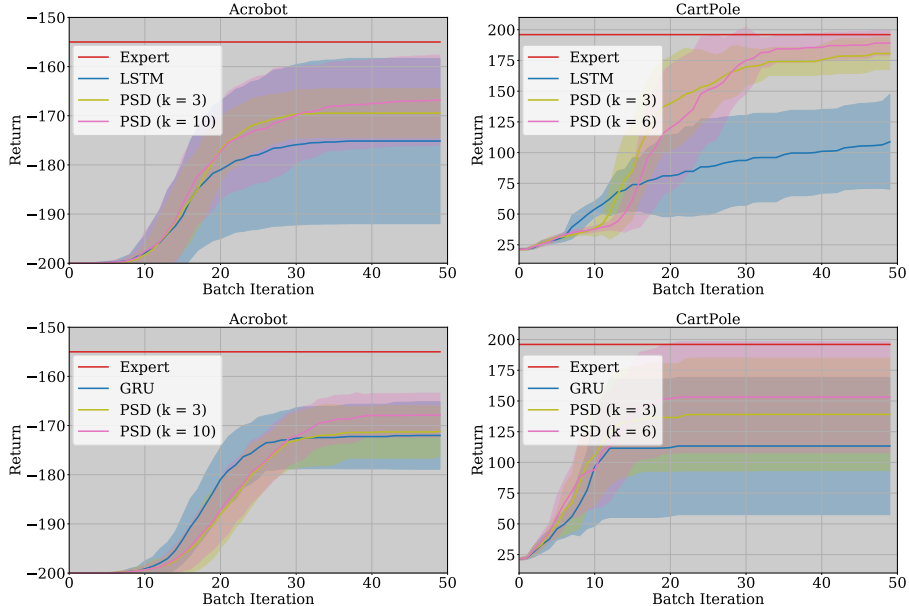

Figure 5: Cumulative rewards for AggreVaTeD and AggreVaTeD+PREDICTIVE-STATE DECODERS on partially observable Acrobot and CartPole with both LSTM cells and GRU cells averaged over 15 runs with different random seeds.

end-to-end filtering performance through the PSD joint objective $\min_{f,F} \mathcal{L} + \lambda \mathcal{R}$. Our experimental results are shown in Fig. 4. The experiments were run with $\phi$ as the identity, capturing statistics representing the first moment. We tested $\phi$ as second-order statistics and found while the performance improved over the baseline, it was outperformed by the first moment. In all environments, a dataset was collected using a preset control policy. In the Pendulum experiments, we predict the pendulum's angle $\theta$. The LQR controlled Helicopter experiments [3] use a noisy state as the observation, and the Hopper dataset was generated using the OpenAI simulation [12] with robust policy optimization algorithm [36] as the controller.

We test each environment with Tensorflow's built-in GRU and LSTM cells [1]. We sweep over various $k$ and $\lambda$ hyperparameters and present the average results and standard deviations from runs with different random seeds. Fig. 4 baselines are recurrent models equivalent to PSDs with $\lambda = 0$.

## 4.2 Imitation Learning

We experiment with the partially observable CartPole and Acrobot domains[3] from OpenAI Gym [12]. We applied the method of AggreVaTeD [46], a policy-gradient method, to train our expert models. AggreVaTeD uses access to a cost-to-go oracle in order to train a policy that is sensitive to the value of the expert's actions, providing an advantage over behavior cloning IL approaches. The experts have access to the *full* state of the robots, unlike the learned recurrent policies.

We tune the parameters of LSTM and GRU agents (e.g., learning rate, number of internal units) and afterwards only tune $\lambda$ for PSDs. In Fig. 5, we observe that PSDs improve performance for both GRU- and LSTM-based agents and increasing the predictive-state horizon $k$ yields better results. Notably, PSDs achieves 73% relative improvement over baseline LSTM and 42% over GRU on Cartpole. Difference random seeds were used. The cumulative reward of the current best policy is shown.

## 4.3 Reinforcement Learning

Reinforcement learning (RL) increases the problem complexity from imitation learning by removing expert guidance. The latent state of the system is heavily influenced by the RL agent itself and changes as the policy improves. We use [18]'s implementation of TRPO [41], a Natural Policy

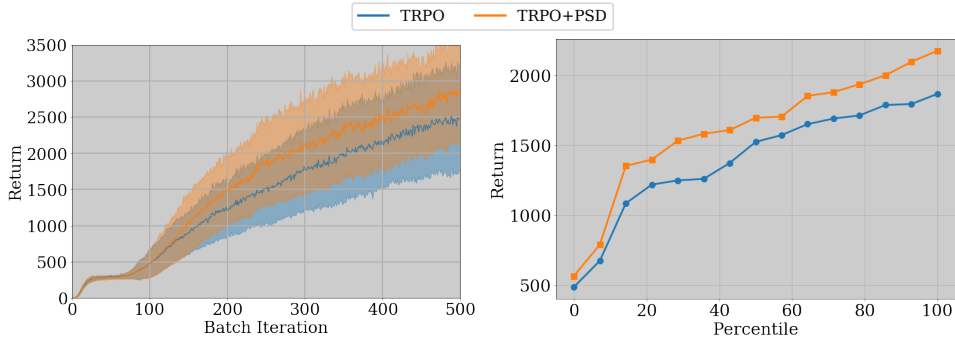

Figure 6: Walker Cumulative Rewards and Sorted Percentiles. $N = 15$, $5e4$ TRPO steps per iteration.

Table 1: *Top:* Mean Average Returns $\pm$ one standard deviation, with $N = 15$ for Walker2d[†] and $N = 30$ otherwise. *Bottom:* Relative improvement of on the means. [*] indicates $p < 0.05$ and [**] indicates $p < 0.005$ on Wilcoxon's signed-rank test for significance of improvement. All runs computed with $5e3$ transitions per iteration, except Walker2d[†], with $5e4$.

|  | Swimmer | HalfCheetah | Hopper | Walker2d | Walker2d[†] |
|---|---|---|---|---|---|
| [41] | $91.3 \pm 25.5$ | $330 \pm 158$ | $1103 \pm 264$ | $383 \pm 96$ | $1396 \pm 396$ |
| [41]+PSDs | $\mathbf{97.0 \pm 19.4}$ | $\mathbf{372 \pm 143}$ | $\mathbf{1195 \pm 272}$ | $\mathbf{416 \pm 88}$ | $\mathbf{1611 \pm 436}$ |
| Rel. $\Delta$ | $6.30\%^*$ | $13.0\%^*$ | $9.06\%^*$ | $8.59\%^*$ | $15.4\%^{**}$ |

Gradient method [28]. Although [41] defines a KL-constraint on policy parameters that affect actions, our implementation of PSDs introduces parameters (those of the decoder) that are unaffected by the constraint, as the decoder does not directly govern the agent's actions.

In these experiments, results are highly stochastic due to both environment randomness and non-deterministic parallelization of `rllab` [18]. We therefore repeat each experiment at least 15 times with paired random seeds. We use $k = 2$ for most experiments ($k = 4$ for Hopper), the identity featurization for $\phi$, and vary $\lambda$ in $\left\{10^1, 10^0, \ldots, 10^{-6}\right\}$, and employ the LSTM cell and other default parameters of TRPO. We report the same metric as [18]: per-TRPO batch average return across learning iterations. Additionally, we report per-run performance by plotting the sorted average TRPO batch returns (each item is a number representing a method's performance for a single seed).

Figs. 6 and 7 demonstrate that our method generally produces higher-quality results than the baseline. These results are further summarized by their means and stds. in Table 1. In Figure 6, 40% of our method's models are better than the best baseline model. In Figure 7c, 25% of our method's models are better than the second-best (98[th] percentile) baseline model. We compare various RNN cells in Table 2, and find our method can improve Basic (linear + `tanh` nonlinearity), GRU, and LSTM RNNs, and usually reduces the performance variance. We used Tensorflow [1] and passed both the "hidden" and "cell" components of an LSTM's internal state to the decoder. We also conducted preliminary additional experiments with second order featurization ($\phi(x) = [x, \text{vec}(xx^T)]$). Corresponding to Tab. 2, column 1 for the inverted pendulum, second order features yielded $861 \pm 41$, a $4.9\%$ improvement in the mean and a large reduction in variance.

## 5 Conclusion

We introduced a theoretically-motivated method for improving the training of RNNs. Our method stems from previous literature that assigns statistical meaning to a learner's internal state for modelling latent state of the data-generating processes. Our approach uses the objective in PSIMs and applies it to more complicated recurrent models such as LSTMs and GRUs and to objectives beyond probabilistic filtering such as imitation and reinforcement learning. We show that our straightforward method improves performance across all domains with which we experimented.

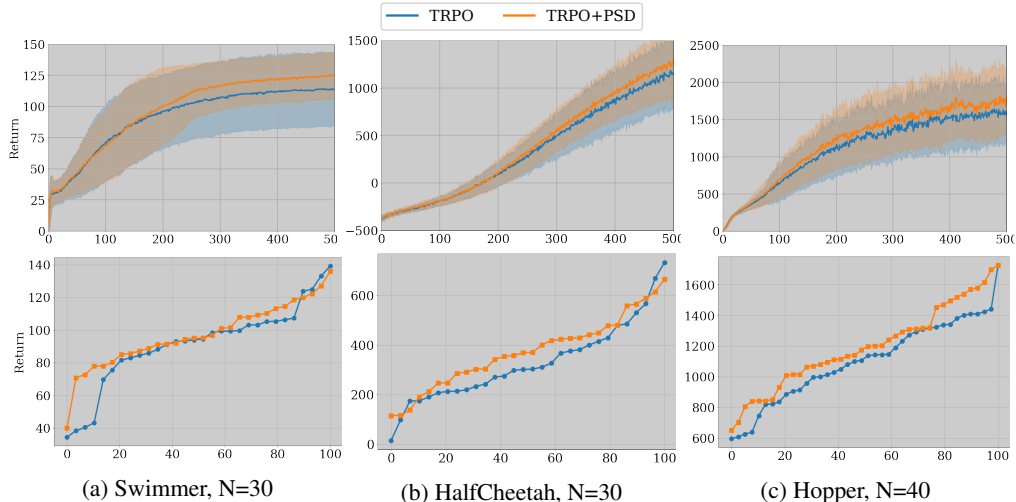

(a) Swimmer, N=30    (b) HalfCheetah, N=30    (c) Hopper, N=40

Figure 7: *Top:* Per-iteration average returns for TRPO and TRPO+PREDICTIVE-STATE DECODERS vs. batch iteration, with $5e3$ steps per iteration. *Bottom:* Sorted per-run mean (across iterations) average returns. Our method generally produces better models.

Table 2: Variations of RNN units. Mean Average Returns $\pm$ one standard deviation, with $N = 20$. $1e3$ transitions per iteration are used. Our method can improve each recurrent unit we tested.

|  | InvertedPendulum | | | Swimmer | | |
|---|---|---|---|---|---|---|
|  | Basic | GRU | LSTM | Basic | GRU | LSTM |
| [41] | $\mathbf{820 \pm 139}$ | $673 \pm 268$ | $640 \pm 265$ | $66.0 \pm 21.4$ | $64.6 \pm 55.3$ | $56.5 \pm 23.8$ |
| [41]+PSDs | $820 \pm 118$ | $\mathbf{782 \pm 183}$ | $\mathbf{784 \pm 215}$ | $\mathbf{71.4 \pm 26.9}$ | $\mathbf{75.1 \pm 28.8}$ | $\mathbf{61.0 \pm 23.8}$ |
| Rel. $\Delta$ | $-0.08\%$ | $20.4\%$ | $22.6\%$ | $8.21\%$ | $16.1\%$ | $7.94\%$ |

## Acknowledgements

This material is based upon work supported in part by: Office of Naval Research (ONR) contract N000141512365, and National Science Foundation NRI award number 1637758.

## Footnotes

[2] In Markov Decision Processes (MDPs), $P(s_{t+1}|s_t)$ may depend on an action taken at $s_t$.

[3]The observation function only provides positional information (including joint angles), excluding velocities.

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
