[Reviews · NeurIPS 2017]

Reviewer 1



The paper proposes an additional loss term for training a RNN that encourages it to capture the latent state. This simple reconstruction loss does help performance on variety of tasks. The main message of the paper is that if future observations are predictable from an internal state, then it means it has captured the latent dynamic of the environment. Such an internal state then should be more useful for the actual task. The claim is supported by diverse set of experiments, although the improvements in the RL tasks are not significant. But here are my concerns: - Since \phi is only defined in sec 4.1, I assume identity is used in the all experiments. Although it is motivated by a more general framework, the models in the experiment are simply doing frame prediction, which is not very novel. There are many works that do future frame prediction both in video and RL. I think the paper will be more clear if the introduction has a paragraph explaining what the model is actually end-up doing: predicting future frames. - Although PSR have nice theoretical guarantees, I don't think they would translate to the proposed model. It would still suffer from non-convexity as the base model RNN is non-convex. - Since there are many works that also perform future frame prediction, the paper needs a "related work" section, instead of spending so much space on explaining RNNs. Other comments: - Sec 4.2 needs more details. It is not clear what is the task, and exactly how the models are trained. [L237] If only the best policy is plotted, why there are error bars in Fig 5? - L246: what "parallelization" means here? - What "5k" and "50k" means in table 1? - What "simple" means in table 2?

Reviewer 2



The authors present a simple method for regularizing recurrent neural networks (specifically applied to low-dimensional control tasks trained by filtering, imitation, and reinforcement learning) that is inspired by the literature on Predictive State Representations (PSRs). In PSRs, processes with latent states are modeled by directly identifying the latent state with a representation of the sufficient statistics of future observations. This stands in contrast to RNNs, which use an opaque hidden state that is optimized by backpropagation to make the whole system model the target of interest through evolution of the hidden state. They propose a simple regularization method for RNNs inspired by PSRs, called Predictive State Decoders, which maintains the parametric form of a standard RNN, but augments the training objective to make the hidden state more directly predict statistics of distant future observations. While direct training of an RNN by backpropagation implicitly encourages the internal state to be predictive of the future, the authors demonstrate that this explicit additional regularization term gives significant improvements on several control tasks. The idea specifically is to augment the standard training loss for an RNN controller (either for filtering, imitation learning, or reinforcement learning) with an additional term that maps the hidden state at each timestep through an additional learned parametric function F, and penalizes the difference between that and some statistics of up to 10 future observations. This encourages the internal state of the RNN to more directly capture the future distribution of observations without over-fitting to next-step prediction, and architecturally improves gradient flow through the network similar to skip-connections, attention, or the intermediate classifiers added in FitNets or Inception. I like the main idea of the paper. It is simple, well-motivated, and connects RNN-based models to a large body of theoretically-grounded research in modeling latent state processes, of which the authors do a good literature review. The experiments and analysis are thorough, and demonstrate significant improvements over the baseline on many of the tasks in the OpenAI gym. The baselines use strong, well-known techniques and optimization methods for imitation learning and RL. I have some concerns that the paper does not offer enough detail about the exact architectures used. For the filtering experiment, it is stated that phi is the identity function, which I assume is true for the future experiments. Additionally, unless I am missing something, the exact functional form of F, the decoder, is never mentioned in the text. Is this a feedforward neural network? Given that the predicted states can sometimes be 10 times the size of the observation dimension, have the authors thought about how to apply this method to models with very large dimensional observations? More detail would be required on exactly the form of F in a camera-ready. Overall, I think this is a well-executed paper that demonstrates a nice synergy between the PSR and RNN methods for latent state process modeling. The method seems simple to implement and convincingly improves over the baselines.

Reviewer 3



The paper proposes a regularization technique of RNNs that tries to predict future inputs from the current hidden state. First of all, I think the paper is written in a very confusing manner. The underlying idea is very simple and intuitive, but it is dressed up in an unconvincing theoretical motivation. You have to read to page 5 in order to finally find out the main idea. The PSR connection is a fine side observation but I don't feel that it a strong motivator. The idea should and can be explained in other, more intuitive terms (e.g. self-supervision) on page 1 itself. But this paper confuses with irrelevant terms and definitions in order to maintain a "theoretical connection". I like the idea itself - this fits in with the self-supervision trend. Predicting future inputs as another objective to train hidden states is intuitive. Basically you are adding a language modeling cost to the RNN. The experiments are decent. I'm surprised that the variance of the PSD is often just as large as without the additional loss (e.g. TRPO), which is contrary to the Unsupervised Auxiliary Tasks paper by Jaderberg et al. 2016. This suggests that the this regularizer is not as strong as other self-supervision tasks. A comparison with the PSR task and other self-supervision tasks would be valuable. Right now, it's hard to tell if this is worth incorporating into self-supervised RL models. Also, it would be very useful to see ablation studies, i.e. the full table for varying the hyperparameter "k". This would help characterize the how well these models can predict future, and when noise starts dominating. To conclude, good simple idea, lacks important ablation experiments, confusingly written.